# Orchestrating Stress Responses in Multiple Sclerosis: A Role for Astrocytic IFNγ Signaling

**DOI:** 10.3390/ijms25147524

**Published:** 2024-07-09

**Authors:** Maria L. Habean, Kaitlin E. Kaiser, Jessica L. Williams

**Affiliations:** 1Department of Neurosciences, Case Western Reserve University, Cleveland, OH 44106, USA; habeanm@ccf.org; 2Department of Neurosciences, Lerner Research Institute, Cleveland Clinic, 9500 Euclid Avenue/NC30, Cleveland, OH 44195, USA

**Keywords:** multiple sclerosis, astrocytes, interferon-gamma, reactive oxygen species, cell death

## Abstract

Multiple sclerosis (MS) is an inflammatory and neurodegenerative disease that is characterized by the infiltration of peripheral immune cells into the central nervous system (CNS), secretion of inflammatory factors, demyelination, and axonal degeneration. Inflammatory mediators such as cytokines alter cellular function and activate resident CNS cells, including astrocytes. Notably, interferon (IFN)γ is a prominent pleiotropic cytokine involved in MS that contributes to disease pathogenesis. Astrocytes are dynamic cells that respond to changes in the cellular microenvironment and are highly responsive to many cytokines, including IFNγ. Throughout the course of MS, intrinsic cell stress is initiated in response to inflammation, which can impact the pathology. It is known that cell stress is pronounced during MS; however, the specific mechanisms relating IFNγ signaling to cell stress responses in astrocytes are still under investigation. This review will highlight the current literature regarding the impact of IFNγ signaling alone and in combination with other immune mediators on astrocyte synthesis of free oxygen radicals and cell death, and cover what is understood regarding astrocytic mitochondrial dysfunction and endoplasmic reticulum stress.

## 1. Introduction

Multiple sclerosis (MS) pathogenesis consists of a series of neuroinflammatory events culminating in central nervous system (CNS) immune cell infiltration, cytokine secretion, demyelination, and axonal degeneration [1,2,3,4]. Due to the complexity of MS pathology, a singular, definitive cause is unknown; however, recent studies have begun to implicate resident CNS cells, such as astrocytes, as key mediators of pathology and disease-associated neurodegeneration [5,6]. Astrocytes are critical support cells for the CNS and are highly responsive to changes in the microenvironment [7,8,9]. Cell stress is a prominent feature of both neuroinflammation and neurodegeneration, and the multifaceted nature of astrocytes makes them a critical CNS regulator of MS pathology. 

A prominent family of cytokines known to alter MS pathology and astrocyte function are interferons (IFNs), which consist of three subtypes: type I (IFNα/β), type II (IFNγ), and type III (IFNλ). Initially studied as a component of the antiviral response, IFNs mediate protection during infection by activating leukocytes and increasing antigen presentation. Within the context of MS, IFNβ has been used as a first-line treatment; however, growing evidence suggests that type II IFNs may also play a beneficial role during chronic stages of disease [10,11,12,13,14]. Pleotropic in nature, IFNγ signaling has been shown to have both proinflammatory and protective functions in a variety of cell types, including astrocytes [15,16], making it important to understand its role in many contexts. Due to the wide scope of IFNγ signaling on various cells in the CNS, we chose to focus on IFNγ signaling in astrocytes during cell stress. Interestingly, aspects of IFNγ signaling in astrocytes include the upregulation of chemokine production, which may contribute to immune cell infiltration [7,9], while simultaneously dampening inflammation through the upregulation of antioxidant proteins and immune checkpoint molecules [13,14]. 

In addition to its role in immune function, IFNγ is also able to initiate additional transcriptional pathways, including cell stress responses [17]. Cell stressors, such as the synthesis of oxygen free radicals, mitochondrial dysfunction, endoplasmic reticulum stress (ERS), and the initiation of cell death and survival, are prominent components of the pathology of many neurological diseases, as reviewed by Maldonado et al. [18]. The primary function of these responses is to return a cell to homeostasis; however, the prolonged activation of stress pathways, as seen during MS, can be detrimental [19,20,21,22,23]. Notably, the effect of IFNγ on these pathways has been well characterized in neurons, oligodendrocytes, and microglia [24,25,26,27]. Cell stress responses, such as oxidative species production and the initiation of cell death pathways, can have dichotomous functions in cells of the CNS depending on the concentration of IFNγ present [28,29,30,31,32,33]. Additionally, other response pathways, like ERS, have been shown to decrease oligodendrocyte viability [27,34,35]; however, it can also protect against excitotoxicity and reduce apoptosis in neurons [36]. While the mechanisms of IFNγ signaling and its impact in modulating cell stress responses in other CNS cell types have been characterized, its role in astrocytes during neuroinflammatory disease is less understood [37,38]. While other reviews comprehensively cover cell stress in other CNS cell types during disease onset or progression, we consider major cell stress pathways, such as oxidative species production and the initiation of cell death in astrocytes, and the effect of IFNγ signaling on these pathways which, to our knowledge, has not previously been covered. 

## 2. Multiple Sclerosis, Astrocytes, and IFNγ Signaling

Epidemiological data within the last five years estimate that nearly 3 million people are currently diagnosed with MS [39]. Approximately 85% of patients diagnosed with MS are first afflicted with relapsing remitting MS (RRMS) in which patients experience recurrent periods of neurological dysfunction followed by partial or complete recovery. As more CNS lesions develop, there is a subtle shift to secondary progressive MS (SPMS) in which there is increasing neurological disability independent of relapses [40]. This shift is thought to result from concurrent axonal and neuronal degradation [41], which is theorized to occur when reaching a pathological threshold for which the CNS is unable to compensate [42]. Approximately 15% of patients experience primary progressive MS (PPMS), a more insidious form of MS in which there is consistent neurological dysfunction with no periods of remission [43]. Current immunomodulatory therapies to alleviate MS symptoms are highly effective at preventing relapses during the early stages of neuroinflammation associated with RRMS [13]. However, they offer limited efficacy for patients with SPMS or PPMS [44,45], where inflammation is thought to be relatively dampened while neurodegeneration progresses [46]. 

Astrocytes are the most prominent glial cell population in the CNS. These important cells have been described for over a century, from Rudolf Virchow’s first description of glial cells [47] to Santiago Ramón y Cajal’s detailing of their connections and basic morphology [48,49]. Deriving from the subventricular zone of the CNS, mature astrocytes are known to provide neurons with functional support, modulate the blood–brain barrier (BBB), and regulate ionic balance [6,50]. In addition, astrocytes assist with neurotransmitter reuptake and the modulation of the synaptic environment [6]. The recent literature describes discrete and unique heterogeneous populations of astrocytes that exist throughout the brain and the spinal cord [51,52]. During injury, hypoxia, and neurodegenerative or demyelinating pathology, astrocytes were previously thought to enter a “reactive” state in which astrocytes change transcriptionally, morphologically, and metabolically [53]. Indeed, astrocytes reside on a spectrum of activated states ranging from homeostatic to reactive, as described by a recent consortium [53]. Moreover, astrocyte morphology does not always correspond to a functional phenotype or impact on surrounding cells, resulting in complex cellular dynamics during disease pathogenesis [53]. This complexity can be observed in MS, where astrocytes have a dynamic role in contributing to both pro- and anti-inflammatory mechanisms. Studies using experimental autoimmune encephalomyelitis (EAE), a commonly used animal model of MS, have uncovered many functions for astrocytes [54]. The mechanisms by which astrocytes exert their beneficial or deleterious functions during neuroinflammation have been a recent topic of study, with the scope of this review focusing on recent work linking the intricacies of IFNγ signaling to cell stress mechanisms. 

The role of IFNγ within the context of MS is complex. Early studies suggested a heavily deleterious role for IFNγ, as it was detected in active MS lesions [55] and elevated in IFNγ-expressing lymphocytes [56]. Early EAE studies reported elevated levels of *Ifng* mRNA within the cerebrospinal fluid during severe clinical disease [57]. IFNγ injections in the spinal cords of rats also resulted in severe inflammation that followed a similar pattern to EAE [58]. However, more recent studies have shown that during chronic EAE, IFNγ plays a neuroprotective role, resulting in reduced demyelination, neurodegeneration, and diminished astrocyte activation [59]. This paradoxical role for IFNγ signaling in astrocytes has recently been corroborated by many other reports [13,14,29,37,60]. The ability of IFNγ to impact how astrocytes either mitigate or advance cellular stress has been investigated to further elucidate how MS progresses and determine the therapeutic implications. 

## 3. Synthesis of Oxygen Free Radicals

The mitochondrial respiratory chain is resistant to small fluctuations in the enzymatic rate and oxidative species production; however, consistent reductions in its activity can increase oxidant production, resulting in pathogenesis [21,61,62]. Oxygen free radicals, such as reactive oxygen species (ROS) and reactive nitrogen species (RNS), are generated in metabolically active organelles for proper biological function [63,64]. During homeostasis, ROS and RNS are important for maintenance and act as alarmin signals during inflammation and cell signaling through pathways important for cell proliferation, differentiation, migration, and angiogenesis [63,64,65,66,67,68,69,70,71,72,73,74]. Electron-displacing reactions are endogenous sources of ROS and RNS [61]. In short, ROS are primarily produced within the mitochondria as a byproduct of ATP formation from the electron transport chain and subsequent reactions such as the Fenton or Heber–Weiss reaction [61,75]. Several enzymes, such as superoxide dismutase (SOD), catalase, and glutathione peroxidase, exist to keep ROS production in check and prevent major perturbations in homeostasis [61,75]. Within peroxisomes, ROS production mainly occurs through the beta-oxidation of fatty acids [73], utilizing enzymes such as acetyl-coenzyme A oxidases. RNS are generated by nitric oxide synthases (NOSs), of which there are three isoforms, neuronal, endothelial, and inducible NOSs, primarily producing nitric oxide (NO), peroxynitrite, nitrogen dioxide, and others [62,64,75,76]. 

Cytokines such as IFNγ have been linked to altered ROS/RNS production and activities of the associated enzymes [77,78]. Specifically, IFNγ signaling is required for iNOS expression in the CNS during EAE [78]. Cells with an astrocyte-like morphology exhibited an enhanced expression of mitochondrial protein and density in active and chronic active MS lesions [79] in addition to the defective expression of proteins associated with mitochondrial respiratory chain complexes, specifically cyclooxygenase 1, in astrocytes, axons, and oligodendrocytes [79,80]. Additionally, the postmortem brains of MS patients exhibit an increased astrocytic NOS activity [81,82]. Together, these data suggest a mechanistic link between IFNγ signaling and ROS/RNS production and a potential role in MS pathology, as the overproduction of ROS and RNS during inflammation can alter astrocyte responses and contribute to pathogenesis and neurodegeneration in MS [21,62,79,80]. A majority of the literature surrounding oxidative stress has focused on other CNS cell types [21,83]; thus, relatively little is known about the mechanisms by which ROS/RNS activate neuroinflammatory pathways in astrocytes, which are still being uncovered.

### 3.1. Detrimental Functions of IFNγ Signaling

Astrocytic ROS and RNS have both intrinsic and extrinsic effects that impact the surrounding cells. Varying concentrations (10 ng/mL–100 ng/mL) of IFNγ determine the intrinsic effects on the cellular stress responses in astrocytes [84]. High doses (100 ng/mL) of IFNγ decrease astrocyte glutamate uptake (14–19% clearance), resulting in the inhibited secretion of neuroprotectants such as thiols and lactates [85], which might contribute to altered neuron physiology. Additionally, IFNγ signaling can cause increases in NOS (65 pmol of citrulline/min/mg or ~400% protein increase) and subsequent NO release (350 ng/mg protein) [81,86,87]. Canonically, IFNγ signals through the receptor tyrosine kinase Janus Kinase (JAK)1 and signal transducer and activator of transcription 1 (STAT1) to activate the transcription factor interferon regulatory factor (IRF)1 [19,84]. This induces the increased expression of NOS, which leads to enhanced RNS [61]. Within astrocytes, combinatorial treatments of IFNγ and interleukin (IL)-1β or tumor necrosis factor (TNF)α result in increased ROS (190% change over control), SOD2 (14-fold change), NOS (1000-fold change), and concomitant decreases in catalase (17-fold change) [29,59,88]. These changes to ROS pathway components results in a buildup of hydrogen peroxide as the catalase activity is dampened. The buildup of ROS following IFNγ and IL-1β stimulation also increases lipid peroxidation (1700 pg/mL) and decreases glutamate uptake (64% of control) by astrocytes, which can contribute to neurotoxicity during MS and EAE [88]. Interestingly, IL-4 exerts similar effects on glutamate uptake when given alongside IFNγ in vitro, contributing to pathology [85]. This illustrates that the intrinsic oxidant production by astrocytes following IFNγ signaling can have significant impact on the viability of neurons, thus contributing to the neurodegeneration seen in MS during neuroinflammation. Besides IL-1β, combined signals from IFNγ and lipopolysaccharide (LPS) induced the NADPH oxidase activity and increased NO production (150 nmoles/mg protein), glucose uptake (28–34 μmol/mg protein), and lactate secretion (53–70 μmol/mg protein), which induced mitochondrial dysfunction and neurotoxicity [86,89,90]. Moreover, neuroinflammation is enhanced by the ROS-induced secretion of IL-6 and TNFα from astrocytes via the activation of JAK/STAT pathways [87,88,90]. Importantly, astrocyte mutants of phagocyte oxidase or the use of an NOS inhibitor ameliorated the detrimental effects of IFNγ and LPS treatment and reduced the secretion of NO and proinflammatory cytokines [89,90]. These studies indicate that IFNγ signaling alters NO production; however, the stimulation of astrocytes with ILs or LPS and IFNγ activates inflammatory pathways and negatively affects neighboring cells, resulting in neurotoxicity, neurodegeneration, and oligodendrocyte death (Figure 1). 

### 3.2. Protective Functions of IFNγ Signaling

In contrast to the detrimental effects of high concentrations of IFNγ, lower levels can lead to the activation of neuroprotective mechanisms in astrocytes. In vitro, low doses of IFNγ (10 ng/mL) can promote astrocyte glutamate clearance by secreting thiols, lactates, and other neuroprotectants, reducing neurotoxicity [85]. In conjunction, low concentrations (0–20 U/mL) of IFNγ signaling in astrocytes results in low ROS (30% change from control) and superoxide production (2-fold change) and increased SOD2 (1.5-fold change), NOS activity, and NO release (175% of control) [81,86,88]. Additionally, endogenous astrocytic NO production following IFNγ and LPS stimulation decreases superoxide, preserves the mitochondrial membrane potential, and decreases mitochondrial swelling following peroxide exposure [89]. This suggests that there are protective mechanisms downstream of IFNγ signaling in astrocytes that work to decrease ROS production and increase NO production. Furthermore, IFNγ induced the immunoproteasome as a possible mechanism for decreased ROS production in astrocytes during EAE [13]. This was substantiated by observing an increased immunoproteasome expression in areas with decreased oxidative stress in astrocytes of chronic active MS lesions [13]. Furthermore, astrocyte-specific IFNγ receptor-deficient mice have significantly exacerbated EAE, specifically during the chronic stages of disease compared to controls, indicating that the protective functions of IFNγ signaling occur during periods of dampened IFNγ production [13,14,59]. Taken together, when evaluating the specific mechanisms initiated by IFNγ in astrocytes, the disease stage, concentration, and context should be considered (Figure 1). 

## 4. Initiation of Cell Death and Survival Mechanisms

Many cell types have reserve mechanisms in place to aid in preventing or postponing apoptosis to allow for time to overcome perturbations. However, the initiation of cell death mechanisms occurs when a cell is unable to return to homeostasis and proper functioning. Cell death is a heavily regulated mechanism that can result in different types of death including apoptosis (active), necrosis (passive), pyroptosis (inflammatory), and ferroptosis (iron dependent) [91,92]. Although there are a variety of cell death pathways, for the purposes of this review, we will focus on apoptosis. Apoptosis is caspase-mediated and highly controlled cell death that can be initiated via both intrinsic and extrinsic pathways [93]. It is an orderly and silent form of cell death characterized by morphological changes in the plasma membrane and nucleus without the induction of inflammation. The intrinsic activation pathway requires the presence of mitochondrial outer membrane permeabilization and the activation of proapoptotic proteins, an event inhibited by anti-apoptotic members of the B cell leukemia 2 (Bcl-2) family [93,94,95]. Cytochrome c is released into the cytosol from the mitochondria following membrane permeabilization and binds to apoptotic protease activating factor-1, eliciting a conformational change and the formation of the apoptosome, a complex primarily composed of caspases. The extrinsic activation pathway can be initiated through four major receptor/ligand pairs [93,96,97]. The receptor/ligand pairs are members of the TNF receptor superfamily, consisting of Fas/Fas ligand (FasL), TNF-related apoptosis-inducing ligand (TRAIL), TNF receptor/TNFα, and programmed death (PD)-1/PD-ligand 1 (PD-L1). The activation of any of these extrinsic pathways results in a similar chain of events consisting of a signaling complex that will cleave initiator caspases and, in turn, activate effector caspases, inducing apoptosis. Contrary to the negative effects of apoptosis, this is an important cellular process that is critical for the formation and function of most systems in the body during development and throughout life [98,99,100,101,102,103]. 

Highly inflammatory events can initiate apoptotic pathways due to elevated cell stress, which can perpetuate the pathology. There is evidence supporting extrinsic pathways of apoptotic activation during MS, as the expression of Fas/FasL and PD-1/PD-L1 occurs on CNS glia [14,104]. Indeed, studies have shown that high levels of IFNγ (100 U/mL) result in the induction of apoptosis in CNS resident cells [32,105,106]. Comparatively, the use of transgenic mice with transcriptionally lower levels of *Ifng* in the CNS does not cause demyelination and oligodendrocyte death, astrogliosis, or microgliosis in murine models of demyelination [107]. Therefore, elucidating the modulatory mechanisms of apoptosis in relation to cytokine signaling can be beneficial for understanding its role in MS pathology. 

### 4.1. The Intrinsic Detrimental and Protective Functions of IFNγ Signaling

Astrocytes are highly adaptive to changes in the surrounding microenvironment, particularly in response to cytokines during neuroinflammation. In vitro, high concentrations (100 ng/mL) of IFNγ are linked to increased astrocyte death [85]. Interestingly, lower concentrations of IFNγ (10 ng/mL) do not induce apoptosis [13,85] unless paired with the pharmacological inhibition of antioxidative pathways, such as the formation of the immunoproteosome [13]. IFNγ levels are relatively high in the CNS during MS relapse and peak clinical EAE, when there is a large influx of activated peripheral immune cells, suggesting that astrocyte viability is likely to decrease during those periods [84]. Additionally, astrocyte pretreatment with IFNγ induced Fas receptor expression, making astrocytes more susceptible to Fas-mediated apoptosis, and resulted in membrane deterioration [108]. Furthermore, astrocytes treated with IFNγ, TNFα, and LPS had increased RNS (~35 μM increase) that correlated with the level of apoptosis [109]. Notably, RNS-induced apoptosis was found to be caspase-dependent and was inhibited by iNOS repression [109]. This is consistent with previous studies demonstrating that the overproduction of oxygen free radicals can induce apoptosis [110]. Furthermore, IFNγ and LPS can increase astrocyte apoptosis by altering the pro- and anti-apoptotic ratios of Bax/Bcl-2 from 1:1 to 2:1 [111]. Similarly, astrocytic IFNγ signaling has also been shown to upregulate caspases, specifically caspase 11 [106], which can activate caspase 3, an important executioner of apoptosis. Overall, the astrocyte-intrinsic mechanisms of cell death require elevated IFNγ concentrations for the initiation of autonomous apoptosis. However, astrocytes are also capable of initiating cell death in surrounding cells (Figure 2). 

Mice with global and astrocyte-specific deficiencies in IFNγ receptors reveal that IFNγ signaling is detrimental during the onset of EAE [9,59]; however, astrocytic IFNγ signaling is protective during more chronic stages [13,14,59]. During chronic EAE, a loss of astrocyte IFNγ signaling resulted in qualitatively enhanced demyelination and axonal degeneration compared to controls [59]. This adds to the mounting evidence for a stage- and concentration-specific effect of IFNγ on astrocytes during neuroinflammation. An example of this is the IFNγ-mediated upregulation of the immunoproteasome in astrocytes. The pharmacological inhibition of the immunoproteasome during EAE results in exacerbated disease with increased demyelination (measured as less MBP^+^ area) [13]. In conjunction, IFNγ receptor-deficient astrocytes exhibited a reduction in immunoproteasome subunit expression (measured as less LMP2^+^ area) and increased demyelination (measured as less MBP^+^ area) that correlated with increased oxidative stress and polyubiquitinated protein accumulation (measured as more PRDX6^+^ and Lys48^+^ areas) [13]. Together these results indicate that the IFNγ-mediated upregulation of the immunoproteasome is critical in protecting cell viability and limiting oxidative species production in the CNS during neuroinflammation. Additional in vitro studies involving a combined treatment of IFNγ (200 U/mL) and IL-1β (10 ng/mL) for 24 h demonstrate no change in apoptosis; however, in the absence of cell death, these astrocytes were still found to upregulate proteins of the extrinsic apoptosis activation pathway, suggesting a role for astrocytic control of apoptosis in the surrounding environment [88]. 

### 4.2. The Extrinsic Detrimental and Protective Functions of IFNγ Signaling

Astrocytes have been shown to increase a variety of cell surface receptors in response to environmental stress that induce apoptosis in neighboring cells. For example, IFNγ signaling in astrocytes induced TRAIL and PD-L1 expression [14,112,113]. Single-cell RNA sequencing identified lysosomal-associated membrane protein-1^+^ and TRAIL^+^ astrocytes that are induced by IFNγ secreted from NK cells to induce T cell apoptosis [113]. Additionally, secreted factors from astrocytes can impact surrounding cells, causing alterations in gene expression and viability. One study evaluating cell viability used conditioned media from astrocytes treated with IFNγ and LPS to determine oligodendrocyte survival. Astrocyte-secreted factors including TNFα and NO induced oligodendrocyte death, which was reversed when the astrocytes were deficient in src-suppressed C-kinase [114]. In contrast, astrocytes and neurons cultured together and treated with IFNγ had reduced neuronal apoptosis compared to neurons cultured alone [85]. This is interesting, as it suggests an anti-inflammatory function for astrocytic IFNγ signaling that is protective for surrounding cells and is exacerbated by LPS. Additionally, the secretion of IL-6 from IFNγ-treated astrocytes mediates neuronal protection even in the presence of neuroinflammatory mediators such as LPS or concanavalin A [37]. Furthermore, the alteration in apoptotic ligands on astrocytes can drastically alter the EAE outcome. EAE in astrocyte-specific FasL-deficient mice was significantly attenuated, suggesting that astrocytic FasL may induce apoptosis in activated, infiltrating T cells during pathogenesis [115]. In addition to FasL, the upregulation of PD-L1 on astrocytes following IFNγ stimulation induced apoptosis in co-cultured leukocytes [14]. The loss of IFNγ receptors on astrocytes during chronic EAE decreased PD-L1 on astrocytes (measured as PD-1^+^ area) and resulted in increased CNS-infiltrating leukocytes, particularly myeloid cells (measured as Iba1^+^/CD45^+^ area), and exacerbated disease [14,116]. Additionally, IFNγ signaling during acute EAE upregulates PD-L1 on astrocytes to interact with microglial PD-1 to attenuate disease progression [116]. This suggests that astrocytes may mitigate CNS inflammation during EAE through apoptosis potentiation or cell exhaustion in infiltrating immune cells (Figure 2).

## 5. IFNγ-Mediated Changes in Cellular Metabolism during MS

### 5.1. Mitochondrial Dysregulation

Mitochondria are double-membraned organelles that are prominently known for their involvement in cellular metabolism via the Krebs cycle as well as oxidative phosphorylation [117]. Mitochondria are distinct in that they have a two-fold expression of both nuclear and maternally inherited mitochondrial DNA [118]. Within the context of MS, mitochondria are vital for ATP production, ROS generation, lipid metabolism, calcium storage, cellular signaling, and the initiation of apoptosis [22,119]. Mitochondrial dysfunction has become a focus of neuroinflammatory and neurodegenerative research, with mounting evidence, a fraction of which is detailed below, that mitochondrial damage correlates with the progression and severity of MS.

In elucidating the complex mechanisms that drive MS progression, changes in cellular metabolism have been investigated as potential drivers of disease progression. Metabolomic profiling of cerebrospinal fluid in an acute EAE model in rats showed distinct changes in metabolites between the onset and peak stages of disease, suggesting that there may be significant changes in cellular metabolism over the course of neuroinflammation [120]. Additionally, when therapeutically delivered during EAE, a mitochondria-targeted antioxidant, MitoQ, suppressed various cytokines, including IFNγ, and reduced neuronal loss in the spinal cords of mice with chronic EAE when given both 10 days before MOG immunization (100 nmol/mouse, i.p. injection) and over the course of EAE (30 days, twice per week) [121]. These findings bolster previous work, wherein dysfunction in homeostatic mitochondrial activity, resulting in decreased ATP production during periods of increased metabolic demand, induces axonal degeneration, which is thought to contribute to the progressive neuronal loss seen in SPMS and PPMS patients [122]. Furthermore, plasma from MS patients revealed significantly higher levels of markers indicating oxidative/nitrative damage, such as carbonyl groups (+1.47 nmol/mg) and 3-nitrotyrosine (+8.36 nmol/mg). These products are theorized to disrupt cellular mitochondrial function in a cascade-type sequence by both reducing ATP production and transportation as well as contribute to mitochondrial gene dysregulation. These irregular processes consequentially contribute to the production of defective, electron-permeable mitochondria, which is thought to further contribute to oxidative and metabolic injury to neurons during MS [123,124]. 

More recent work has provided greater insights into the connection between astrocytic mitochondrial dysfunction in the progression of MS and how astrocytes may drive progression as well as be a target for novel therapeutics. Sadeghian et al. observed that during the onset of murine EAE, astrocytes upregulate glycolysis via the increased expression of the enzyme phosphofructokinase-2, observed visually via immunohistological imaging of GFAP^+^/PFK-2 astrocytes, suggesting early metabolic changes in astrocytes during disease that could be targeted [125]. In addition, astrocytes in inflamed areas of EAE tissues did not express the mitochondrial dye Mitotracker Green, indicating that astrocytic mitochondria could be depolarized, a condition that is attributed to the onset of mitochondrial damage or dysfunction [125,126]. Moreover, lactate, an energy-rich metabolite released by astrocytes to provide metabolic support to neurons [127], can be diminished when glycolytic pathways are disrupted. Certain genes upregulated in EAE related to metabolic sphingolipid production, such as *B4galt6*, a lactosylceramide synthase, contribute to disease progression. This was shown via ameliorated chronic progressive EAE in mice with B4GALT6 suppression with PDMP treatment starting at day 40 (20 mg/kg, i.p.), while exacerbated EAE was observed in a separate cohort of mice given lactosylceramide (10 µg, i.p.) every 3 days starting at day 35 [128]. In addition, astrocytes with inhibited B4GALT6 did not respond to LPS/IFNγ stimulation (100 ng/mL); however, with exogenous lactosylceramide administration meant to mimic autocrine-type signaling, astrocytes displayed a marked increase in their response to LPS/IFNγ treatment. Finally, an increased expression of B4GALT6 (8.26-fold change) and lactosylceramide was seen in MS patient lesions, suggesting that the balance of pro- and anti-inflammatory lipid production and metabolism can impact disease progression [128]. Pharmacological studies have used Miglustat, a drug classically used to treat dysregulated enzymatic lipid breakdown in Gaucher and Niemann–Pick disease. When Miglustat was given by gavage (600 mg/kg) before the chronic stage of EAE (~20 days post-induction), it suppressed pathogenic astrocyte metabolism and ameliorated EAE by limiting the production of TNFα and IFNγ. Further, Miglustat restored lactate production via the suppression of lactosylceramide generation and cytosolic phospholipase A2 activation [129]. Overall, astrocytic mitochondria are known to be altered during EAE; however, the exact mechanisms are currently not fully understood.

Recently, astrocytic metabolic reprogramming throughout EAE was investigated on a large scale through transcriptomic analysis, which validated that astrocytes upregulate genes involved in glycolysis, the tricarboxylic acid cycle, and glycogen degradation [130]. The upregulation of these pathways correlated with a morphological transition to a more active state in lesion-associated cerebellar astrocytes, suggesting that a shift in metabolism is concurrent with a reactive phenotype. This may perpetuate disease pathogenesis and could be explored as a source of potential therapeutic targets. Furthermore, astrocyte-induced pluripotent stem cells derived from MS patients had altered expressions of mitochondrial-related genes, which correlated with increased fission and decreased mitochondrial mass [131]. To gain a deeper understanding of the metabolic dynamics of astrocytes during MS, investigators further assessed the metabolomic profile of astrocytes and found that mitochondria have increased proton leakage, electron transport capacity, and mitochondrial uncoupling [131]. In addition, astrocytes from active MS lesions exhibited an upregulation of respiratory electron transport and cell stress pathways compared to controls [131]. Together, this further validates the unique metabolic profile exhibited by astrocytes in response to the changes in metabolic demands brought on by MS pathology.

The expression of transcription co-factor proliferator-activated receptor gamma coactivator 1-α (PGC-1α) and downstream mitochondrial antioxidant enzymes peroxiredoxin-3 and thioredoxin-2 are increased in reactive GFAP^+^ astrocytes within active MS lesions, shown via immunohistochemistry. The increases seen are theorized to act as a protective mechanism for astrocytes against oxidative stress from ROS production and prevent ROS-induced neuronal death when comparing neurons cocultured with lentiviral-induced PGC-1α^+^ astrocytes vs. neurons with mock-transduced astrocytes [132]. Indeed, astrocyte overexpression of PGC-1α and its downstream targets lowered both the proinflammatory gene expression and protein secretion of IL-6 and CCL2 as well as reduced exogenous IFNγ/TNFα-induced ROS production [132]. These cytokine-associated changes provided early evidence for the protective potential of astrocytes during metabolic changes in MS. More recent studies investigated a novel role for liver kinase B1 (LKB1) in astrocytes during chronic EAE using an astrocyte-specific conditional knockout of LKB1 [133], which has several functions in the CNS, outlined in detail by Kuwako and Okano [134]. Heterozygous deletion of LKB1 in astrocytes was sufficient to alter the expression of mitochondrial electron transport chain complexes and reduce the expressions of metabolic genes during chronic EAE. The CNS of these mice had increased IFNγ^+^IL-17^+^ cells, shown using flow cytometry (1.1% increase), and in vitro, reduced LKB1 expression resulted in upregulated IFNγ target genes such as *Stat1*, *Ciita*, and *H2* and enhanced the disease severity and spinal cord pathology [133]. In addition to the described deleterious effects of IFNγ on astrocyte mitochondrial dysfunction, there is still much to learn about IFNγ signaling in astrocytes during the chronic stages of MS and EAE, where IFNγ has a more protective role [13,14]. 

### 5.2. Endoplasmic Reticulum Stress and the Unfolded Protein Response

The ER, a site of calcium storage, protein synthesis, and lipid metabolism, can become flooded by proteins that require proper folding, inducing the ERS response in an effort to return the organelle to a homeostatic state. Periods of high protein demand, such as neuroinflammatory periods, calcium fluctuations, and cell stress from ROS/RNS production, can induce the ERS response. The ERS activates the unfolded protein response (UPR), which transcriptionally upregulates ER genes to increase the capacity of protein folding, block protein translation for homeostasis restoration, and target misfolded proteins for degradation in the ER. However, prolonged ERS can result in ERS-induced cell death. The UPR is made up of three different pathways: protein kinase RNA-like endoplasmic reticulum kinase (PERK), activating transcription factor 6 (ATF6), and inositol-requiring enzyme 1α (IRE1α), where each activates a separate transcriptional pathway for the return to ER homeostasis. All three UPR pathways result in nuclear translocation and the initiation of transcription of ERS genes to support amino acid metabolism, redox homeostasis, chaperone proteins, lipid synthesis, protein degradation, and apoptosis to reduce ERS. A surplus of inflammatory signals during neuroinflammatory disease can increase protein synthesis and increase ROS/RNS, resulting in oxidatively damaged proteins in CNS cells [13]. Therefore, the ERS can significantly influence the pathology following the activation of the UPR. 

ERS pathways are activated in multiple neurodegenerative and neuroinflammatory diseases, including Alzheimer’s disease, Parkinson’s disease, amyotrophic lateral sclerosis, and MS. Short-term ERS activation is thought to be protective by altering the transcriptome and proteome; however, prolonged ERS initiates inflammatory and apoptotic pathways that can promote neurotoxicity [20]. MS patient studies have shown that ERS markers are upregulated in white matter lesions (3-7-fold change) compared to healthy individuals [135]. Additionally, these ERS indicators were found to be upregulated (20–55% increase) in inflammatory active MS lesions, specifically within glia [136]. These data suggest a link between ERS and UPR to MS pathogenesis downstream of inflammation. During periods of high glutamate, there are increases in glucose-regulated protein 78, a protein necessary for regulating ERS and UPR, which is thought to protect neurons from glutamate toxicity and prevent neuron death during EAE [36,137,138]. Furthermore, the inflammatory environment during MS contributes to ERS induction in glial cells, specifically oligodendrocytes [35]. The mechanisms of inflammation-induced ERS within resident CNS cells is currently under investigation, with a majority of the literature focused on oligodendrocytes. There have been studies, however, investigating ERS initiation within astrocytes outside the context of inflammation that may shed light on inflammatory causes of UPR. 

Mitochondrial dysregulation and ROS/RNS production can initiate ERS due to the increased presence of oxidized proteins [139,140]. Cytokine induction of the ERS and the UPR has been heavily studied within oligodendrocytes during MS and EAE [34,35,141,142,143,144,145]; however, less is known in other CNS cells. Astrocytic ERS is regulated by JAK1/STAT3 through ATF4, suggesting a possible role for cytokines in ERS induction [139]. In fact, IFNγ signaling induces ERS in astrocytes and upregulates IL-6 (600 pg/mL) [146]. Further, PERK inhibition can block increases in cytokine gene expression following ERS initiation in astrocytes [147]. The induction of the ERS in astrocytes has been covered further in other recent reviews [38,148]. 

### 5.3. Cell Death and Survival Pathways

Cell death and survival pathways are understudied in relation to astrocytic IFNγ signaling in MS. With an emergence of new types of cell death pathways, including ferroptosis and pyroptosis, there are many new avenues to uncover the IFNγ-mediated regulation of these mechanisms. While it has been shown that astrocytes are susceptible to ferroptosis during MS and other related neurodegenerative diseases [149,150], the mechanisms governing astrocytic ferroptosis are still unclear. Additionally, there is increasing evidence of inflammasome activation, a mediator of pyroptosis, in astrocytes during EAE [151,152,153,154,155]; however, little is known regarding inflammasome activation via cytokine signaling. Specifically, a greater understanding of how key cytokines prevalent in MS can alter cell stress responses may lead to new possibilities for therapeutic target development.

## 6. Discussion

Understanding how IFNγ controls cellular stress during neuroinflammation in the presence of several other inflammatory factors is important for the potential targeting of specific downstream mechanisms during neuroinflammatory diseases like MS (Figure 3). IFNγ signaling has both detrimental and protective functions that become prevalent at specific concentrations and stages of disease. Cell stress responses are important during neuroinflammation and are intrinsically beneficial; however, the dysregulation of these processes can result in detrimental outcomes.

On a cellular level, astrocytic function is dependent on the concentration of IFNγ present in the cellular microenvironment. In a vacuum, higher dosages of IFNγ (100 ng/mL) evoke a more reactive, proinflammatory astrocyte phenotype that exacerbates deleterious pathways and downstream mediators related to cellular stress that subsequently contribute to heightened disease pathogenesis. Further exacerbation of these dysregulated processes occurs when astrocytes are surrounded by other proinflammatory cytokines, as is seen during disease onset and progression. However, lower dosages of IFNγ (10 ng/mL) paradoxically evokes an activated astrocyte morphology that initiates cellular mechanisms that are protective in conditions that elicit heightened cell stress. These protective mechanisms can be maintained even when other cytokines are present (Figure 1 and Figure 2). These contradictory functions of IFNγ not only demonstrate the complex spectrum of astrocyte activation during disease, but also the importance of concentration and time. It is understood how IFNγ secretion from immune cells impacts the environment during the pathology; however, specifics regarding the extra- and intra-cellular changes leading to increased cell stress and how cells of the CNS, including astrocytes, respond remain to be uncovered. While the immune component of MS has been thoroughly investigated [156], there is now a shift in focus to the intrinsic cellular mechanisms of resident CNS cells to further understand the pathology.

## 7. Conclusions

Previously, IFNγ was thought to be solely detrimental during MS; however, recent research is shedding light on the protective mechanisms induced by IFNγ signaling during chronic disease, a timepoint where MS therapies are critically lacking. In many contexts, IFNγ signaling is concentration dependent, whereby excessive IFNγ, as seen during acute MS and EAE, is detrimental to astrocyte function, while lower IFNγ levels during chronic disease attenuates inflammation. As a cell that exhibits a wide array of functions, astrocytes can greatly alter the pathological outcome by modulating the activation of stress response pathways depending on the levels of IFNγ that are present in the microenvironment in conjunction with other cytokines and cellular products generated during disease. A deeper appreciation of the duality of inflammatory signaling and the resultant astrocyte response during pathogenic cell stress may lead to a shift in the view of neuroinflammation and its implications in the context of neurodegeneration and repair. Utilizing the protective nature of downstream targets and pathways may inform novel therapeutic modalities for neuroinflammatory diseases like MS.

## Figures and Tables

**Figure 1 ijms-25-07524-f001:**
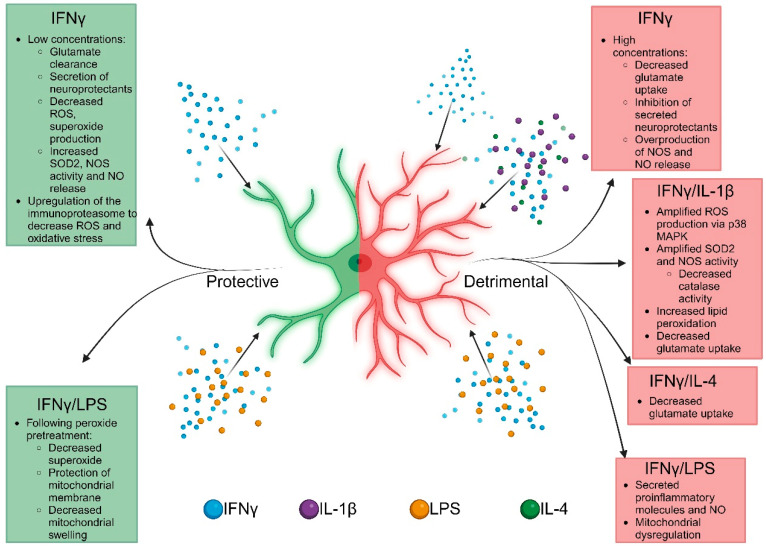
Schematic representation of the downstream regulation of ROS/RNS in astrocytes in response to IFNγ alone or in conjunction with other cytokines. Protective mechanisms are shown in green, while detrimental mechanisms are depicted in red, highlighting the concentration- and context-specific effects of IFNγ during neuroinflammation. Image created using BioRender.com (accessed on 20 May 2024).

**Figure 2 ijms-25-07524-f002:**
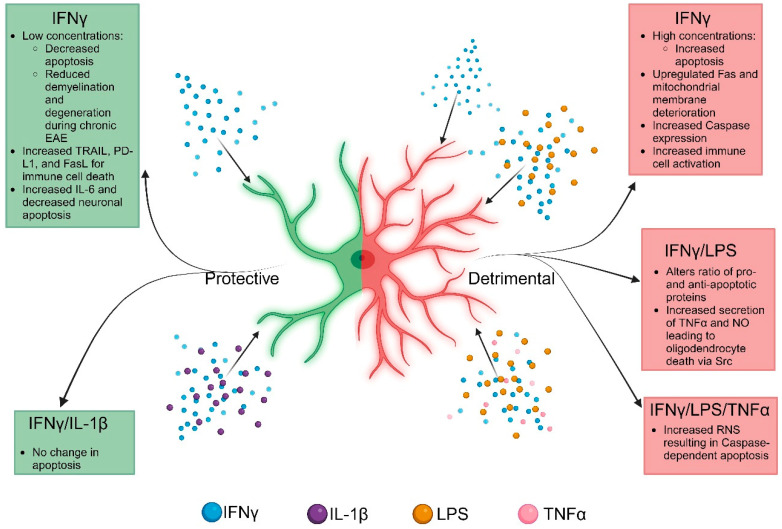
Schematic representation of the regulation of cell death and survival of astrocytes in response to cytokine crosstalk. While the beneficial mechanisms downstream of IFNγ are shown in green, the detrimental responses are depicted in red. Currently, a majority of the literature is centered on apoptosis, with little known about astrocyte survival mechanisms during neuroinflammatory disease. Image created using BioRender.com (accessed on 20 May 2024).

**Figure 3 ijms-25-07524-f003:**
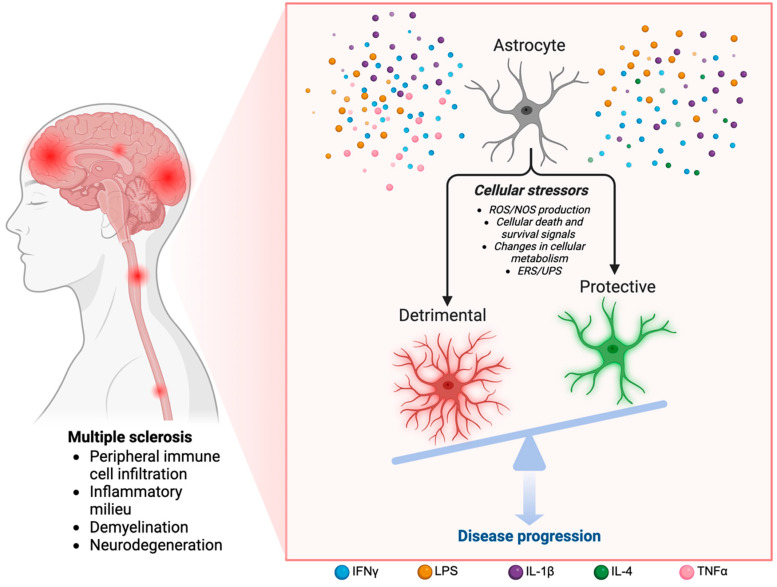
Graphical summary of the effects of cytokine signaling on astrocytes during MS pathogenesis. MS has several pathological hallmarks including immune cell infiltration from the periphery, resulting in an inflammatory milieu, the demyelination of axons, and eventual neurodegeneration. Astrocytes are highly dynamic cells that respond to various inflammatory stressors such as ROS/NOS production, cellular death signals, changes in metabolism, and intrinsic ERS/UPS mechanisms. The timing, concentration, and combination of cytokines during disease can contribute to either a detrimental or protective astrocyte phenotype and how they respond to cell stress. The resultant astrocyte state can either contribute to disease progression or to a neuroreparative environment. Image created using BioRender.com (accessed on 20 May 2024).

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
