# Peer review of "Orchestrating Stress Responses in Multiple Sclerosis: A Role for Astrocytic IFNγ Signaling"

_ijms, 2024, doi:10.3390/ijms25147524_

Round 1

Reviewer 1 Report

Comments and Suggestions for Authors

This is a very well written and balanced review on the impact of IFN-g on astrocytes during the course of EAE and MS. The authors are well known in this field and have previously published a significant number of valuable papers on this topic.

This is a complicated subject because many contradictory results have been obtained regarding the pathophysiological role of IFN-g in multiple sclerosis. It is thus all the more remarkable that the authors managed to depict in detail the different functions carried out by this cytokine in this disease.

 The authors could include in their review a paragraph on the commonly accepted notion of A1 vs A2 astrocytes and how it may or may not fit into the scheme proposed in this review.

Minor comments

The word “prominent” is twice in the abstract in particular.

Ref N°10 is not appropriate as it shows that IFN-g exacerbates MS

Figure 3 legend ends with the 2 words: The resultant. I guess there is something missing here.

Author Response

Comment 1: This is a very well written and balanced review on the impact of IFN-g on astrocytes during the course of EAE and MS. The authors are well known in this field and have previously published a significant number of valuable papers on this topic. This is a complicated subject because many contradictory results have been obtained regarding the pathophysiological role of IFN-g in multiple sclerosis. It is thus all the more remarkable that the authors managed to depict in detail the different functions carried out by this cytokine in this disease.

Response 1: Thank you for the positive feedback; we greatly appreciate it.

Comment 2: The authors could include in their review a paragraph on the commonly accepted notion of A1 vs A2 astrocytes and how it may or may not fit into the scheme proposed in this review.

Response 2: We thank the reviewer for this suggestion. We have added this to the introduction in lines 104-107. 

Comment 3: The word “prominent” is twice in the abstract in particular.

Response 3: Thank you for pointing out this redundancy. We have removed the first “prominent” in the abstract. 

Comment 4: Ref N°10 is not appropriate as it shows that IFN-g exacerbates MS

Response 4: We appreciate the reviewer for pointing this out. We have removed reference 10 from this statement.

Comment 5: Figure 3 legend ends with the 2 words: The resultant. I guess there is something missing here.

Response 5: We have checked the Figure 3 legend for accuracy and appreciate the reviewer for pointing this out. Assuming this was a formatting intricacy, the remainder of the Figure 3 legend was found on line 780 at the end of the manuscript. 

Reviewer 2 Report

Comments and Suggestions for Authors

Introduction

Why is this review important for readers? Are there other reviews like this? Who is the possible audience for this manuscript?

How did the authors select the information for this review?

Synthesis of oxygen free radicals

This section lacks novelty. The information presented here is commonly shown in many manuscript reviews. 

Detrimental functions of IFN gamma signaling

Much information is descriptive and quantitative data is necessary. When authors mention the terms increase or decrease, they should explain in percentage or fold increase or decrease. For instance: "This induces increased expression of NOS which leads to enhanced RNS". How much is the increase? "Varying concentrations of IFN gamma".  "High doses of IFN gamma". The authors should express quantitatively the concentrations and doses reported in each study. This type of correction should be performed throughout the manuscript.

The discussion and conclusion section should be rewritten. It is not clear which are the conclusions of this work. What can the readers obtain from this work for further work on this topic? The authors should mention the perspective of this topic.

Lines 462-472 are general information, this information does not provide a discussion on this topic.

Comments on the Quality of English Language

no comments

Author Response

Comment 1: Introduction: Why is this review important for readers? Are there other reviews like this? Who is the possible audience for this manuscript? How did the authors select the information for this review?

Response 1: Thank you for posing these interesting and important questions. This review is important for readers as it draws attention to a gap in the astrocyte literature specifically with regards to modulation of intracellular cell stress events during IFNg-associated neuroinflammation. We specifically mention this in lines 51-54, 73-76, 111-114, 420-423, and 508-510. There are no other reviews that focus specifically on astrocytic IFNg control of cell stress processes making this manuscript unique. The reason we chose the information for this review can be found in lines 72-76. The audience for this manuscript is the scientific community and those interested in gaining a comprehensive picture of the role of cell stress mechanisms during disease and how they can be modulated by immune mediators. 

Comment 2: Synthesis of oxygen free radicals: This section lacks novelty. The information presented here is commonly shown in many manuscript reviews. 

Response 2: We thank the reviewer for their insight and have altered the section to highlight the gap in knowledge for the contribution of astrocytes to ROS/RNS production. While ROS/RNS is known to contribute during disease and specifically known to have a role in MS, most of the literature is focused on other CNS cell types, specifically macrophages/microglia and oligodendrocytes. Relatively little information is known regarding astrocytic ROS/RNS and how production by these cells can alter disease outcome. This section has been adjusted to reflect this in lines 157-158.

Comment 3: Detrimental functions of IFN gamma signaling: Much information is descriptive and quantitative data is necessary. When authors mention the terms increase or decrease, they should explain in percentage or fold increase or decrease. For instance: "This induces increased expression of NOS which leads to enhanced RNS". How much is the increase? "Varying concentrations of IFN gamma".  "High doses of IFN gamma". The authors should express quantitatively the concentrations and doses reported in each study. This type of correction should be performed throughout the manuscript.

Response 3: We appreciate the reviewer for bringing this to our attention and have added specifics to concentrations and differences observed in outcome measure, where available. This change has been made throughout the manuscript.

Comment 4: The discussion and conclusion section should be rewritten. It is not clear which are the conclusions of this work. What can the readers obtain from this work for further work on this topic? The authors should mention the perspective of this topic.

Response 4: We thank the reviewer for this critique. The discussion and conclusion have been rewritten so that the importance of the research on this topic are clearer. The discussion and conclusion are now separate and have been adjusted to better highlight the perspective of this topic. 

Comment 5: Lines 462-472 are general information, this information does not provide a discussion on this topic.

Response 5: We appreciate this comment and have removed this material.

Round 2

Reviewer 2 Report

Comments and Suggestions for Authors

The lines mentioned in the reply letter do not correspond to the lines in the corrected manuscript. It was difficult to find the changes made by the authors in the manuscript.

How did the authors select the information for this review?

Synthesis of oxygen free radicals: This section lacks novelty. The information presented here is commonly shown in many manuscript reviews. Was this section modified only by two lines?

The discussion and conclusion section should be rewritten. It is not clear which are the conclusions of this work. What can the readers obtain from this work for further work on this topic? The authors should mention the perspective of this topic. What were the changes made by the authors in these sections?

Comments on the Quality of English Language

no comments

Author Response

We thank the reviewer for bringing this to our attention. In our revised manuscript, revisions are highlighted in yellow and page numbers correspond to the .doc file submitted. If the reviewer is referring to the MDPI formatted version, this may have created alternative spacing and thus new line numbers that do not correspond to our submitted .doc file. In parentheses, we have also included the line numbers that correspond to our original revisions using the MDPI formatted manuscript. We have no control over how the submitted .doc file converts to the MDPI formatted manuscript, so we are hopeful that this is helpful to the reviewer in assessing our revisions.

Comment 1: Introduction: Why is this review important for readers? Are there other reviews like this? Who is the possible audience for this manuscript? How did the authors select the information for this review?

Response 1: Thank you for posing these interesting and important questions. This review is important for readers as it draws attention to a gap in the astrocyte literature specifically with regards to modulation of intracellular cell stress events during IFNg-associated neuroinflammation. We specifically mention this in lines 51-53 (MDPI: 43-45), 72-76 (MDPI: 63-67), 111-114 (MDPI: 101-104), 420-423 (MDPI: 414-416), and 505-507 (MDPI: 504-506). There are no other reviews that focus specifically on astrocytic IFNg control of cell stress processes making this manuscript unique. The reason we chose the information for this review can be found in lines 72-76 (MDPI: 63-67). The audience for this manuscript is the scientific community and those interested in gaining a comprehensive picture of the role of cell stress mechanisms during disease and how they can be modulated by immune mediators. 

Comment 2: Synthesis of oxygen free radicals: This section lacks novelty. The information presented here is commonly shown in many manuscript reviews. 

Response 2: We thank the reviewer for their insight and have altered the section to highlight the gap in knowledge for the contribution of astrocytes to ROS/RNS production. While ROS/RNS is known to contribute during disease and specifically known to have a role in MS, most of the literature is focused on other CNS cell types, specifically macrophages/microglia and oligodendrocytes. Relatively little information is known regarding astrocytic ROS/RNS and how production by these cells can alter disease outcome. This gap in knowledge has been highlighted in lines 155-158 (MDPI: 144-147).

Comment 3: The discussion and conclusion section should be rewritten. It is not clear which are the conclusions of this work. What can the readers obtain from this work for further work on this topic? The authors should mention the perspective of this topic. What were the changes made by the authors in these sections?

Response 3: We thank the reviewer for this critique. The discussion and conclusion have been rewritten so that the importance of the research on this topic are clearer. The discussion contains a summary and take-home message of the topic covered and the conclusions are now separate and have been adjusted to better highlight the perspectives. 

Round 3

Reviewer 2 Report

Comments and Suggestions for Authors

The manuscript can be accepted for publication

Comments on the Quality of English Language

no comments